# Basal Unit Radar Characteristics at the Southern Flank of Dome A, East Antarctica

Shuai Yan[1,2], Duncan A. Young[2], Donald D. Blankenship[2], Tyler J. Fudge[1], Duyi Li[2], Laura Lindzey[3], Hunter Reeves[2,4], Alejandra Vega-Gonzalez[2,5], Shivangini Singh[2,4], Megan Kerr[2,4], Emily Wilbur[1], Michelle Koutnik[1]

[1]Department of Earth and Space Sciences, University of Washington, Seattle, 98195, USA
[2]University of Texas Institute for Geophysics, Jackson School of Geosciences, University of Texas at Austin, Austin, 78758, USA
[3]Ocean Engineering Department, Applied Physics Laboratory, University of Washington, Seattle, 98105, USA
[4]Department of Earth and Planetary Sciences, Jackson School of Geosciences, University of Texas at Austin, Austin, 78712, USA
[5]Department of Environmental Sciences, University of Virginia, Charlottesville, 22904, USA

*Correspondence to*: Shuai Yan (syan94@uw.edu)

**Abstract.** The basal unit near the base of the Antarctic Ice Sheet (AIS) plays a critical role in AIS dynamics and the preservation of old ice, yet its structure and origin remain poorly understood. Using a new airborne ice-penetrating radar dataset collected by the NSF Center for Oldest Ice Exploration (NSF COLDEX), we investigate the radar characteristics of the basal unit at the southern flank of Dome A, East Antarctica. We combine manual mapping with Delay-Doppler analysis to characterize the spatial distribution of incoherent scattering and to distinguish between two types of radar-apparent basal unit top boundaries: a sharp transition from specular to scattering reflections (type I) and a gradual disappearance of specular reflections due to radar signal attenuation (type II). We find that incoherent scattering is widespread upstream and decreases downstream, correlating with both subglacial topographic roughness and a shift from type I to type II boundaries. These patterns are interpreted as resulting from spatial variability in englacial temperature, with warmer ice downstream enhancing signal attenuation and obscuring radar features. Although incoherent scattering is not visible in the downstream region, its absence may reflect radar detection limits rather than true absence of scattering reflectors in the basal unit. Moreover, the observed correlation between scattering and subglacial roughness suggests deeper geological controls in which subglacial lithology influences both basal temperature and subglacial geomorphology.

## 1 Introduction

### 1.1 The Antarctic basal unit

Ice-penetrating radar (IPR) has been a foundational tool in advancing our understanding of the cryosphere (Schroeder et al., 2020). IPR data have played a central role in mapping subglacial topography (e.g., Pritchard et al., 2025), characterizing subglacial hydrology (e.g., Livingstone et al., 2022; Yan et al., 2022a), and reconstructing past glacial and environmental changes in polar regions (e.g., Beem et al., 2018; Jamieson et al., 2023). Englacial stratigraphy mapped by IPR provides a

valuable record of past ice sheet dynamics and offers critical guidance for identifying promising sites in the search for old ice cores (Bingham et al., 2025). Near the base of the Antarctic Ice Sheet (AIS), however, radar sounding often encounters a

35 distinct zone—referred to as the basal unit (e.g., Goldberg et al., 2020) or deep scattering zone (e.g., Cavitte, 2017)—where coherent and traceable englacial reflections cannot be detected. The basal unit typically manifests as either an echo-free zone or a zone of non-stratigraphic, incoherent echo (Fig. 1), both indicative of complex and poorly understood basal processes. In addition to its unique radar appearance, studies of ice flow at Little Dome C constrained by the stratigraphic horizons in the upper part of the ice column (Cavitte et al., 2021) indicate that the basal unit there may be largely stagnant, in contrast to the

40 overlying stratigraphic unit, which shows evidence of flow (Chung et al., 2023, 2025). This potential decoupling raises important questions about the physical nature and origin of the basal unit at Little Dome C and elsewhere. Given that observed basal units indicate complex basal processes, they are significant to our understanding of AIS dynamics and the preservation of old ice in Antarctica.

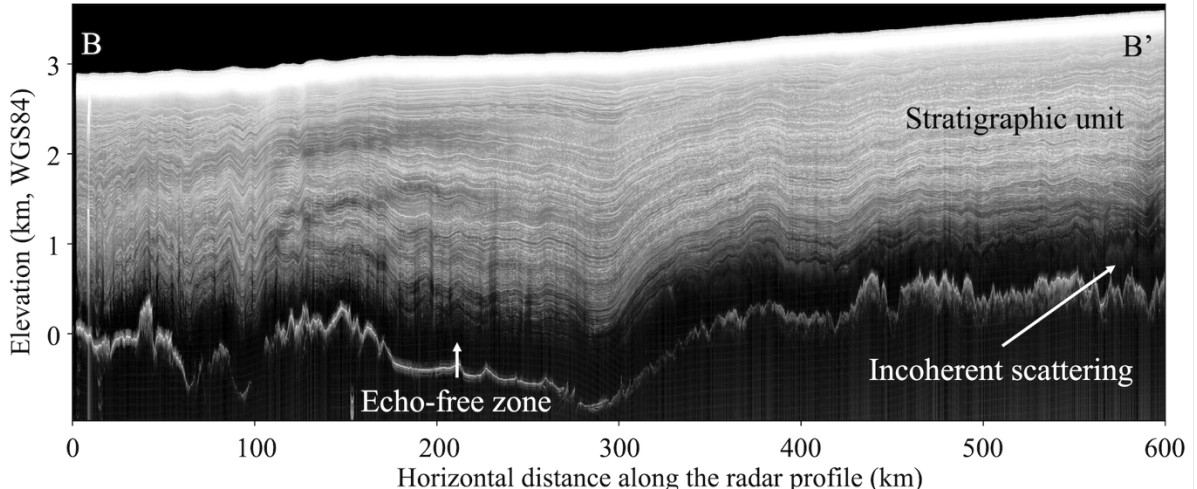

**Figure 1. Example ice-penetrating radargram showing a cross-sectional view of the ice sheet. The location and orientation of this profile are indicated in Fig. 2 as transect B–B′. Radar transect name: CLX/MKB2n/R72a.**

The properties and dynamics of the Antarctic basal unit are poorly constrained, and several mechanisms have been proposed to explain its radar-obscuring character:

    (1) One explanation suggests that radar echoes disappear where dielectric contrast diminishes and where ice core

stratigraphy becomes disrupted due to ice flow–induced deformation (Drews et al., 2009). Under this hypothesis, no special deformation profile is assumed for the basal unit relative to the stratigraphic unit above. Instead, the hypothesized disruption arises from ice flowing over rough subglacial terrain. Because the basal unit is in direct contact with this terrain, it is more susceptible to such disruption.

    (2) Enhanced attenuation near the base of the ice sheet may also contribute to the absence of echo, which is associated

with higher englacial temperature (MacGregor et al., 2015). Once their signal is reduced below the noise floor of

the radar system, englacial reflections can no longer be detected. The ice–bedrock interface typically produces a much stronger reflection and often remains visible, even after experiencing greater attenuation (traveling through a thicker ice column).

(3) Debris entrained during basal freeze-on or introduced through bedrock erosion may also contribute to the incoherent backscatter observed in radar data (Franke et al., 2023, 2024; Winter et al., 2019). It is proposed that such embedded debris, acting as point reflectors, scatters the radar signal and hinders the resolution of internal features within the basal unit. This mechanism has been proposed in regions such as the Gamburtsev Mountains, East Antarctica (Bell et al., 2011), and northern Greenland (Leysinger Vieli et al., 2018), where basal freeze-on processes and rugged subglacial terrain are thought to enhance debris incorporation.

(4) Deformed or folded layering may also contribute to the observed incoherent echo. Wolovick et al. (2014) demonstrated that ice flowing over basal slippery patches can induce large-scale folding. Such folding can disrupt stratigraphic coherence, potentially producing the diffuse scattering signals detected within the basal unit. This deformation-driven mechanism may act in tandem with freeze-on processes (e.g., Bell et al., 2011), where the refreezing of subglacial water and debris entrainment further complicate the radar signature of deep ice.

(5) Other studies point to variations in ice crystal orientation fabric (COF) as a contributing factor (Lilien et al., 2021). By analyzing radar data together with deep ice-core data, Mutter and Holschuh (2025) find that incoherent scattering often coincides with either gradual shifts or rapidly fluctuating COF in deep ice, particularly in regions where strain is localized due to grain size–dependent strength differences. Interestingly, they also note that "macro-scale deformation and layer folding at scales below the range resolution of radar do not seem to result in incoherent scattering or induce an echo-free zone", challenging earlier assumptions.

Collectively, these hypotheses underscore the complexity of basal unit processes and highlight the need for further observational, modeling, and sampling efforts to better characterize this relatively poorly understood part of the ice sheet given the range of hypotheses that can contribute to their character in radar data.

## 1.2 The southern flank of Dome A

This study focuses on the southern flank of Dome A, East Antarctica, a region that remains one of the least studied sectors of the continent despite its glaciological significance (Fig. 2) (Pritchard et al., 2025). Dome A sits atop the subglacial Gamburtsev Mountains, which are believed to have played a central role in the initiation of East Antarctic glaciation (Bo et al., 2009). The geomorphology of the subglacial Gamburtsev Mountains likely records the early history of ice sheet development in this region (Lea et al., 2024). The southern flank, situated between Dome A and the South Pole, is characterized by rugged subglacial topography (Pritchard et al., 2025), an extensive hydrological network (Kerr et al., 2023; Wolovick et al., 2013), and the probable presence of a subglacial sedimentary basin (Aitken et al., 2023). As ice flows from Dome A toward the South Pole, the surface slope of the ice sheet decreases markedly (Fig. 2-a), coinciding with a subtle deflection in flow direction toward the Recovery Subglacial Highlands. Along this flow path, the ice transitions from the

rugged subglacial terrain of the Gamburtsev Mountains to the relatively smooth bedrock of the South Pole Basin farther
downstream (Fig. 2-b). These changes in subglacial conditions and ice flow configuration likely influence basal unit
dynamics. Together, these factors make the southern flank of Dome A an ideal natural laboratory for investigating the
physical properties of the basal unit and the processes governing its formation and variability.

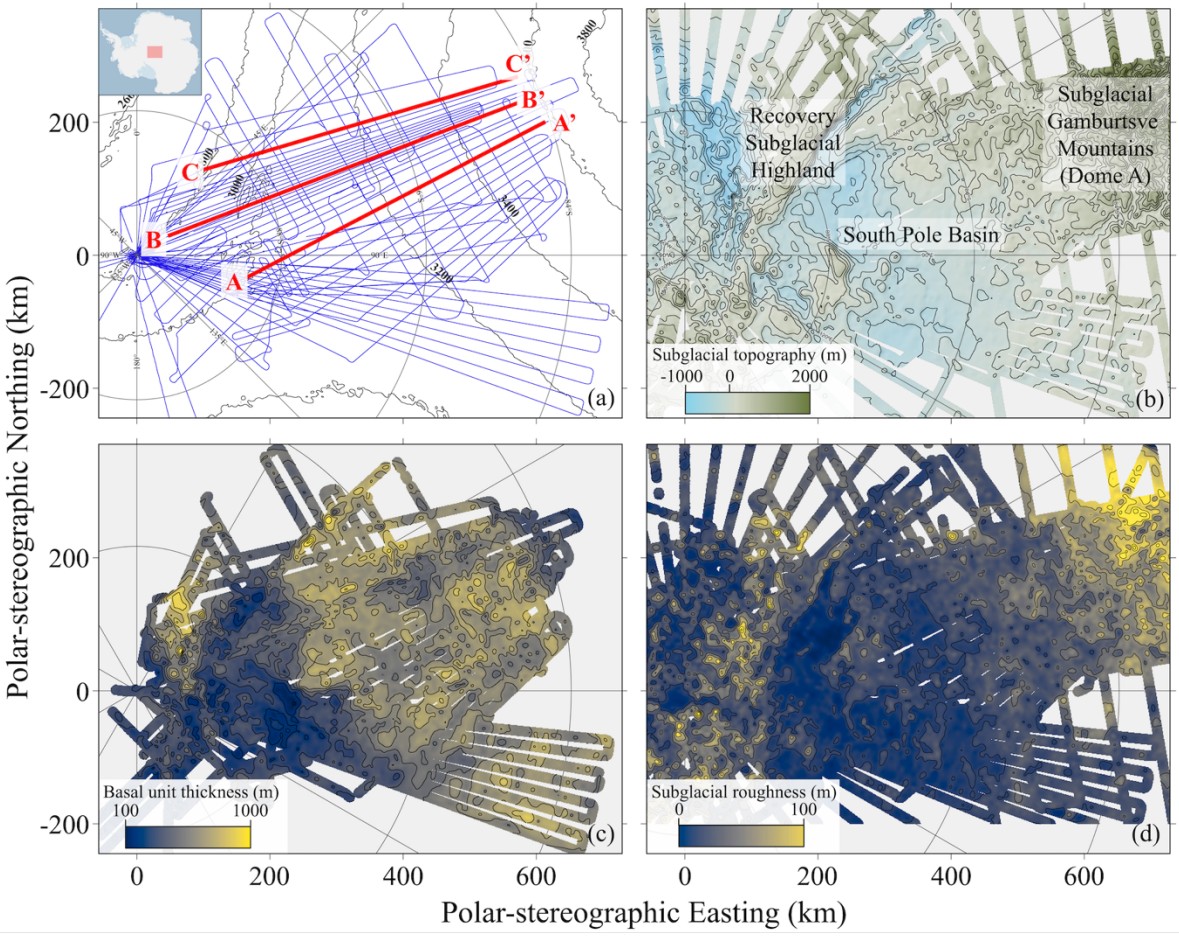

**Figure 2. Data products from the NSF COLDEX airborne geophysical survey. (a) Survey flight lines (blue) overlaid on ice surface
elevation contours at 200 m intervals (black). The location of the survey region is shown in the inset map at upper left. (b)
Subglacial topography of the survey region with 200 m elevation contours. (c) Mapped thickness of the basal unit with 100 m
thickness contours. (d) Subglacial roughness across the survey region, represented as the standard deviation of bed elevation over
a 400 m window, with contours at 20 m intervals. All the maps in this figure, Fig. 6, and Fig. 7 are in the WGS 84 / Antarctic Polar
Stereographic (EPSG:3031) coordinate system.**

The National Science Foundation Center for Oldest Ice Exploration (NSF COLDEX) is commissioned to explore Antarctica
for the oldest continuous ice core, with the goal of advancing our understanding of the evolution and future of Earth's
climate system. As part of this effort, NSF COLDEX coordinated two seasons of airborne geophysical surveys over the
southern flank of Dome A (2022-23 and 2023-24) (Fig. 2-a) (Young et al., 2025). The survey design includes a majority of

flight lines aligned with the overall ice flow direction (from grid north-east to grid south-west in Fig. 2), facilitating future ice flow modeling efforts. Additional lines oriented perpendicular to the flow were included to support across-transect tracing of englacial stratigraphy and leveling of potential field datasets, such as airborne gravity and magnetics. This new airborne geophysical dataset provides new, direct measurements of ice thickness, englacial stratigraphy, and subglacial topography, and offers critical insights into the regional subglacial hydrological and geological conditions. In this study, we leverage this new dataset to investigate the radar characteristics of the basal unit. Specifically, we map the spatial extent and thickness variation of incoherent echo within the basal unit and use Delay-Doppler analysis to investigate the potential mechanisms driving changes in basal unit radar characteristics.

## 2. Methods:

### 2.1 Mapping the presence of incoherent scattering within the basal unit

During the NSF COLDEX airborne geophysics campaign, two independent IPR systems were deployed on the survey aircraft: the MARFA 60 MHz radar system developed by the University of Texas Institute for Geophysics (Young et al., 2016), and a newly developed UHF array based on the University of Kansas accumulation radar (Kaundinya et al., 2024). The new UHF array is designed primarily for high-resolution mapping of englacial radio-stratigraphy in the upper portion of the ice column, but lacks the penetration depth needed to image the full ice thickness or resolve the basal unit. Consequently, this study focuses on data collected with the MARFA system, which is optimized for deep ice penetration and provides enhanced imaging of the lowermost part of the ice sheet.

The boundary between the basal unit and the overlying stratigraphic ice unit, along with the spatial extent and thickness variation of incoherent echo, is manually mapped using the DecisionSpace Geosciences 10ep software package, which contains semi-automatic tracing algorithms and enables cross-transects tracing and comparison (Cavitte et al., 2021; Yan et al., 2025b) (Fig. 3). We define the top of the basal unit—i.e., the boundary between the basal unit and the overlying stratigraphic unit—as the deepest depth at which any clear and traceable englacial reflection is observed. This boundary does not necessarily occur at the same englacial reflection across the study area; rather, it varies spatially, with some deeper reflections visible in certain locations but absent in others. For calculating ice unit thicknesses, a constant velocity of 168.5 m µs⁻¹ is assumed for radio wave propagation in ice. To improve the clarity of englacial reflections during manual tracing, 2-D focusing was applied following the procedure described in Peters et al. (2007), which helps correct for along-track scattering and enhance signal coherence. The resulting thickness map of the basal unit is reported in Yan et al. (2025a) (Fig. 2-c), while this manuscript presents the mapped distribution of incoherent echo within the basal unit.

### 2.2 Delay-Doppler analysis

Delay-Doppler analysis distinguishes between specular and scattering reflections in IPR sounding data. As a radar passes over a target, smooth (typically with roughness less than ⅛ of a wavelength) and continuous surfaces will tend to reflect

incident energy specularly at a defined angle. In contrast, rough surfaces or volume scatterers distribute energy over a broad range of angles. This effect can be seen through the delayed off nadir energy over rough surfaces (Campbell et al., 2013; Oswald and Gogineni, 2008; Young et al., 2016), and can also be detected through along track Doppler filtering (Michaelides and Schroeder, 2019). The phase history of subsurface scatterers enables estimation of the angles at which echoes are returned, providing additional insight into scattering geometry (Peters et al., 2005; Schroeder et al., 2015; Tyler et

al., 1992).

In this study, we apply Doppler filtering using 1000-meter along-track apertures to compare the SNR of returns from three angular windows: nadir and ±11° off-nadir (in air), with evaluations spaced every 500 meters along the flight path. Energy that appears only in one angular view is classified as specular, while that observed in all views is classified as scattering. We use the gradient in the ratio of specular to scattering of 10 dB/µsec to identify the top of the basal unit and a 3 dB

scattered/specular ratio threshold to identify englacial scattering below that limit.

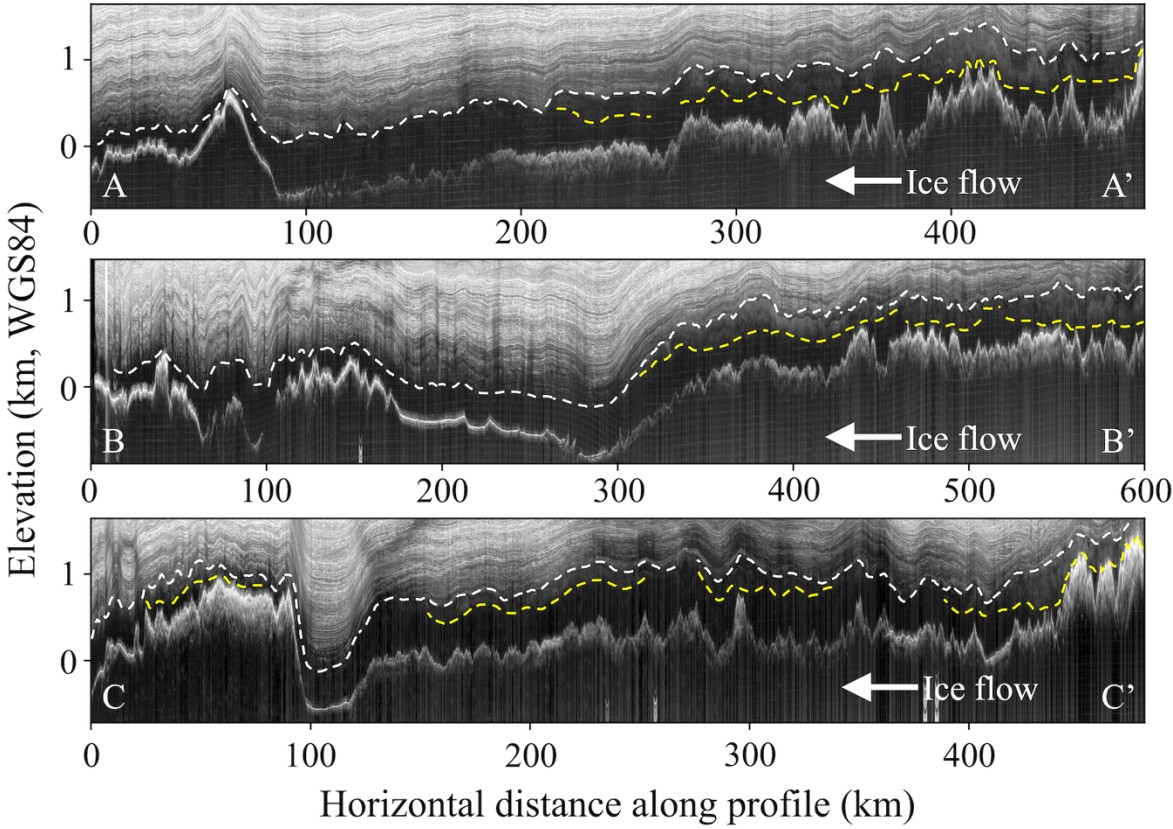

**Figure 3. Three example radargrams showing the presence and thickness variation of incoherent scattering within the basal unit. In each radargram, white dash line marks the top of the basal unit, and yellow dash line marks the bottom of incoherent scattering. The locations and orientations of these profiles are indicated in Fig. 2. Radar transects names: AA':**

**CLX_MKB2n_R56a; BB': CLX/MKB2n/R72a; CC': CLX_MKB2n_R84b.**

## 3. Absence of stratigraphic reflection within the basal unit

Our Delay-Doppler analysis reveals that the majority of internal reflections within the overlying stratigraphic ice unit exhibit predominantly specular characteristics, consistent with coherent and well-preserved dielectric contrasts (Fig. 4). The bed reflection varies between specular and non-specular across the survey region, likely reflecting spatial variability in basal material properties and the presence or absence of subglacial water (Carter et al., 2017). Within the basal unit, non-stratigraphic, incoherent echo is attributed to scattering energy, indicating a shift in the nature of radar reflectors—potentially due to embedded debris, ice fabric heterogeneity, or other complex basal processes.

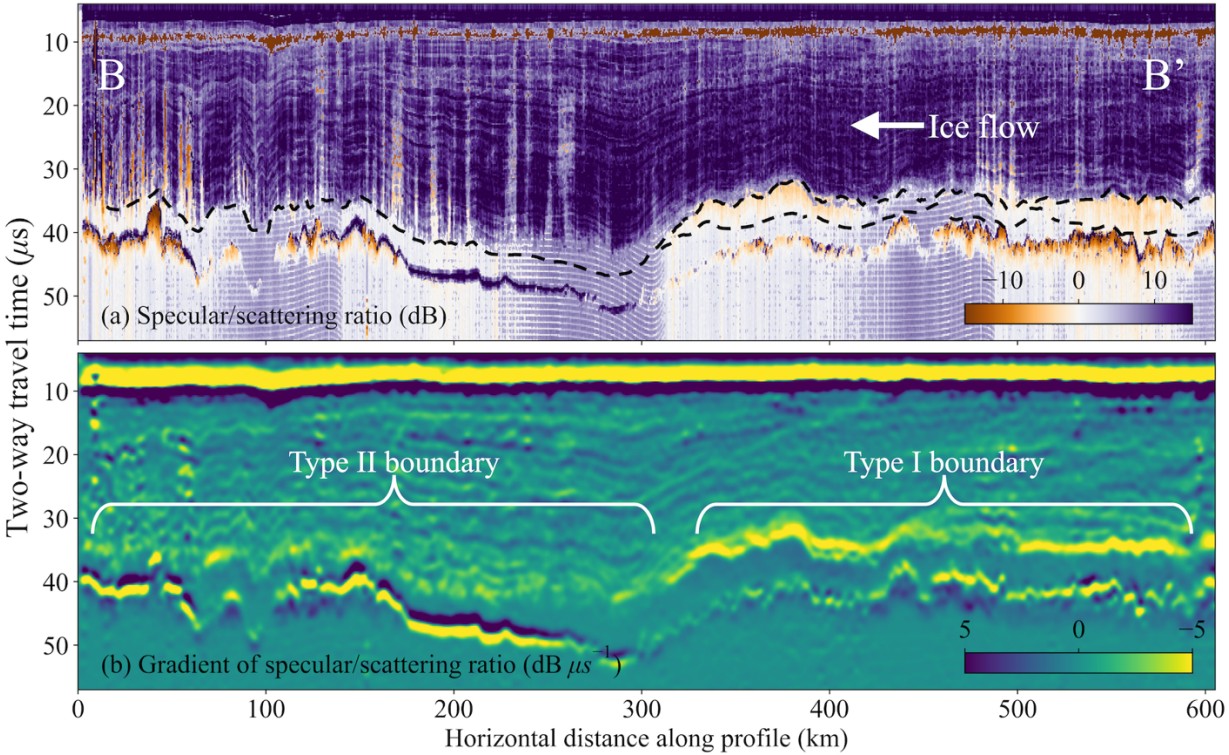

Figure 4. Delay-Doppler analysis for the radar transect shown in Fig. 1 (B–B′ in Fig. 2). (a) Power ratio between specular and scattering reflections, with black dash lines marking the top and bottom of the incoherent scattering echo. (b) Vertical gradient of the power ratio, highlighting the sharpness of transitions.

There are two primary mechanisms for the disappearance of clear, traceable stratigraphic reflections within the basal unit. The first is a change in the nature of reflectors—from specular reflectors in the overlying stratigraphic ice to diffuse, scattering reflectors in the basal unit. The second is enhanced englacial attenuation, which causes both specular and scattering signals near the bed to weaken below the radar system's noise floor and become undetectable. In the first case, we expect a relatively sharp transition from specular to scattering energy; in the second, a gradual decay of specular reflections with depth. In visual identification and manual mapping of basal unit thickness, the top boundary of the radar-apparent basal unit is essentially the shallower of these two depths: either the point of reflector transition (hereafter referred to as a type I

boundary) or the depth at which reflections fade below the noise floor (type II boundary). A conceptual sketch illustrating this distinction is provided in Fig. 5.

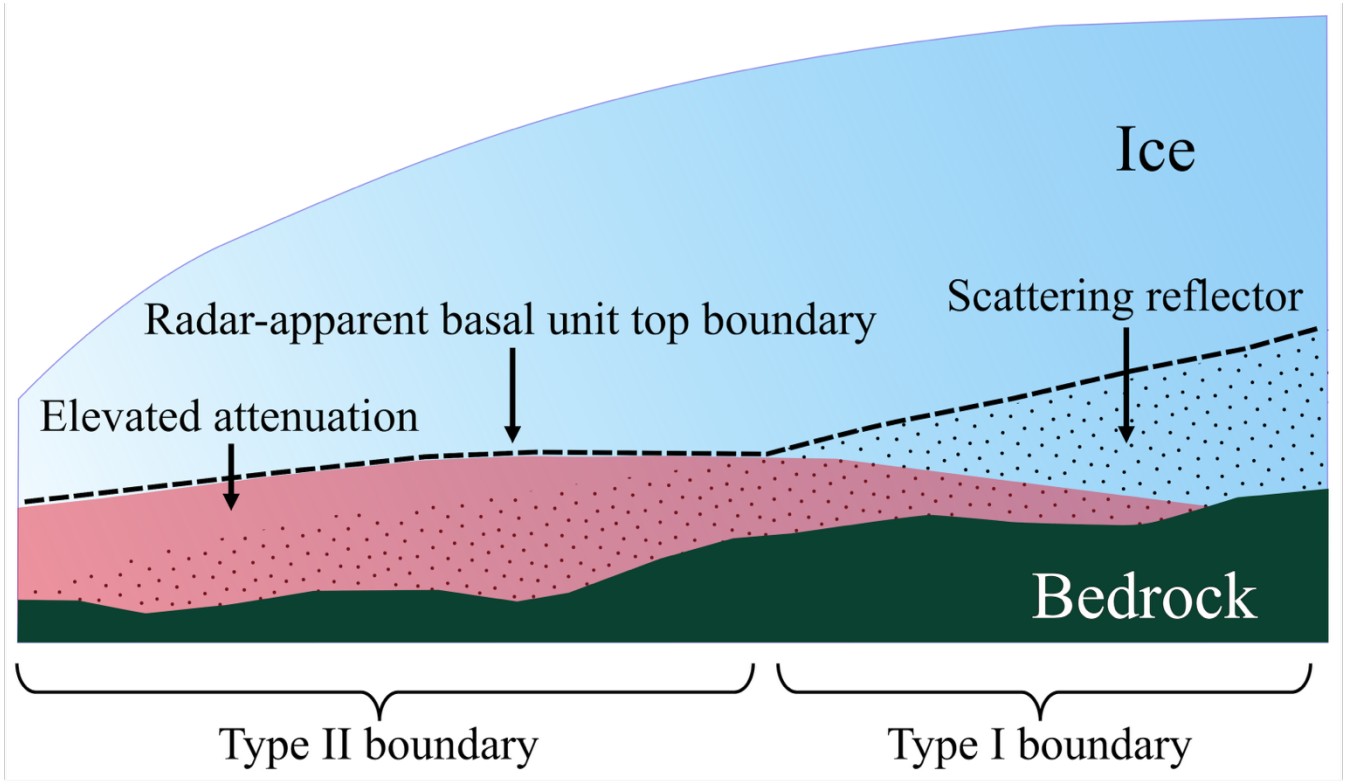

**Figure 5. Conceptual sketch illustrating the distinction between type I and type II basal unit top boundaries. Black dots represent scattering reflectors within the basal unit. The red-shaded region indicates areas of elevated englacial attenuation, where both specular and scattering reflections weaken and fall below the radar system's noise floor. The radar-apparent basal unit boundary is shown as a dashed black line. We note that the variations in ice thickness and subglacial topography shown in this conceptual sketch are intended only as a schematic illustration and do not necessarily correspond to actual correlations between such variations and basal unit boundary types.**

Delay-Doppler analysis can help distinguish between these two types of boundaries. Specifically, we compute the ratio of specular to scattering energy (Fig. 4-a), then calculate the vertical gradient of this ratio over a two-way travel time interval of 1 microsecond (Fig. 4-b). A steeper negative gradient indicates a sharp transition from specular to scattering reflections—consistent with a type I boundary—while a more gradual decline suggests progressive attenuation of specular energy with depth, indicative of a type II boundary. This quantitative approach provides a useful diagnostic for boundary classification, especially in regions where visual interpretation alone may be ambiguous.

Between the southern flank of Dome A and the South Pole, we observe both types of boundary. Our Delay-Doppler analysis suggests that the basal unit boundary is predominantly type I in the upstream region—marked by a sharp transition from specular to scattering reflections—whereas in the downstream region, it is primarily type II, characterized by the gradual fading of specular reflections below the noise floor (Fig. 4, Fig. 6-a). This pattern suggests that, in the downstream region,

where the transition appears more subdued, specular energy may be attenuated below the noise floor at the top of the basal unit, rather than abruptly replaced by scattering.

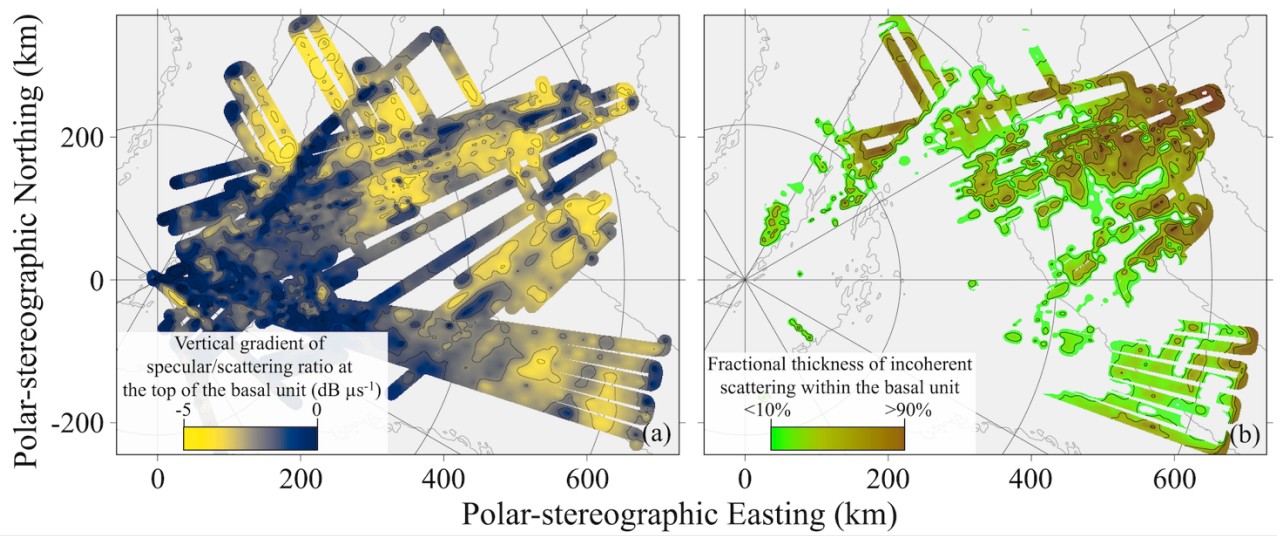

**Figure 6.** Spatial transition from type I to type II radar-apparent basal unit boundaries. (a) Vertical gradient of specular energy at the top of the basal unit, contoured at 0.5 dB µs-1 intervals. (b) Fractional thickness of incoherent scattering within the basal unit, contoured at 20% intervals.

## 4. Presence of incoherent scattering within the basal unit

We observe a widespread presence of incoherent scattering within the basal unit in the upstream portion of the survey region (grid northeast in Fig. 2 and Fig. 6). To quantify its spatial thickness variation, we calculate its fractional thickness relative to the total thickness of the basal unit (Fig. 6-b). Near Dome A (i.e., the upstream area), the basal unit is almost entirely filled with incoherent scattering, with fractional thickness values approaching 100%. This fraction gradually decreases downstream as the ice flows toward the South Pole Basin, and eventually, the incoherent scattering disappears entirely and the basal unit manifests solely as an echo-free zone (Fig. 3, Fig. 6-b). Notably, during this transition from full scattering to entirely echo-free, the scattering consistently diminishes from the base upward—i.e., the echo-free zone first develops at the bottom of the basal unit, immediately above the bedrock, and then progressively thickens upward as it evolves downstream (Fig. 3). Also, the appearance and thickness variation of the incoherent scattering also correlate with the rate at which specular horizons fade vertically, which reflects a transition from type I to type II boundaries (Fig. 6).

The COLDEX survey is situated directly downstream of the Antarctica's Gamburtsev Province (AGAP) Project (Corr et al., 2020). It has been hypothesized that the AGAP IPR sounding reveals packages formed by freezing of subglacial water and subsequent entrainment of debris (Bell et al., 2011; Creyts et al., 2014; Wolovick et al., 2013). We provide side-by-side comparisons of this basal unit as imaged by the COLDEX and AGAP IPR sounding at several intersection points in Fig. 7. We notice that (1) the incoherent scattering exhibits characteristics similar to the unit directly overlying the basal freeze-on

package, and (2) this incoherent scattering is widespread within the AGAP survey in the region intersecting the COLDEX survey, that is, around and downstream of the area where widespread basal freeze-on was inferred by Bell et al. (2011). Based on this observation, we consider the incoherent scattering unlikely to represent the basal freeze-on package given its distinct radar signature. Instead, we interpret the incoherent scattering as arising from either (1) deformation and folding caused by ice flowing across slippery patches of the bedrock (as suggested for other locations by Wolovick et al., 2014), or

(2) variations in ice crystal orientation fabric (as suggested for other locations by Mutter and Holschuh, 2025).

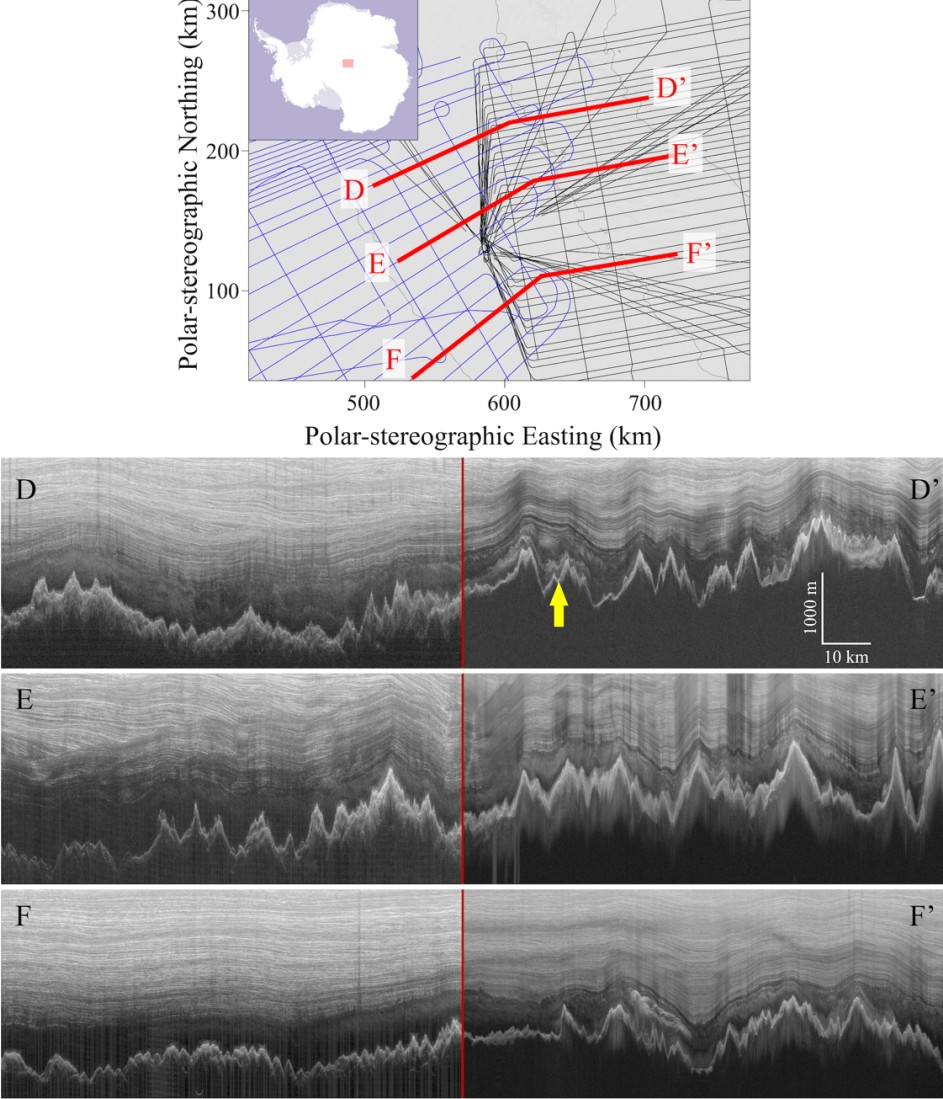

**Figure 7. Side-to-side comparison of the COLDEX survey and the AGAP survey at three of their intersection points. The yellow arrow highlights an example of the basal freeze-on packages as hypothesized by Bell et al., 2011. The location of the survey region is shown in the inset map at upper left. We note that the radar system used in the AGAP survey operates at a different center frequency (150 MHz), which results in different vertical resolution and may alter the appearance of the same reflector—particularly for reflectors whose characteristic dimensions are comparable to the radar wavelength.**

We interpret both the observed variation in incoherent scattering thickness and the shift from type I to type II boundary types as potential indicators of spatial heterogeneity in englacial temperature. In particular, we suggest that warmer ice in the downstream region leads to increased radar signal attenuation, which reduces the detectability of deep reflections and obscures the specular-to-scattering transition. As a result, the radar-apparent basal unit thickness may reflect not only physical changes in ice properties but also thermal conditions influencing signal propagation. Additionally, subglacial melting in warmer areas may remove scattering reflectors from the base of the basal unit, thereby shifting the remaining scattering reflectors to greater depths, while simultaneously raising the critical depth at which radar reflections fall below the noise floor. Together, these effects contribute to a transition from type I to type II boundary. This interpretation also aligns with the observation that widespread subglacial lakes are found in the inner South Pole Basin, near the South Pole point, while almost no subglacial lakes are detected in the outer South Pole Basin, closer to Dome A (Kerr et al., 2023). Within this conceptual framework, the downstream extent of the scattering reflectors remains uncertain. In the downstream area, incoherent scattering disappears within the basal unit, but this does not necessarily indicate that reflectors are absent. Instead, they may simply be undetectable due to increased radar attenuation in warmer ice.

There are alternative explanations for the observed decline of incoherent scattering downstream. If the scattering arises from disturbed or folded stratigraphy, or formed during basal freeze-on, the dielectric contrasts responsible for scattering may be reduced as the ice is advected downstream. Two processes in particular— diffusion and ice deformation—can diminish these contrasts over time. Diffusion acts to smooth out electrical property variations, reducing the amplitude of dielectric contrasts and thereby weakening the radar-scattering signal. This process becomes increasingly effective with time and distance along the flowline, especially near the bed, where ice temperatures are higher and diffusion rates are enhanced (Fudge et al., 2024). In parallel, mechanical deformation can further homogenize the ice and reduce the amplitude of contrasts. This deformation is also likely strongest near the ice-rock interface. Together, diffusion and deformation may progressively erase the dielectric contrasts responsible for the scattering echo, leading to its gradual disappearance downstream.

The radar data we have so far cannot definitively resolve the causes of (1) the absence of stratigraphic reflections and (2) the presence and thickness variation of incoherent scattering within the basal unit. To resolve these uncertainties and test the outstanding hypotheses, future work should prioritize targeted coring campaigns and in situ borehole observations, particularly in zones where radar data show a transition from incoherent scattering to echo-free conditions. Platforms such as RAID (Goodge et al., 2021; Shackleton et al., 2025) may provide access to these challenging depths with relatively high drilling speed and efficiency. Additionally, polarimetric radar sounding can provide valuable insight into variations in crystal orientation fabric (COF), which may further constrain these hypotheses (Hills et al., 2025). In parallel, numerical modeling will be essential. Future simulations could quantify spatial patterns of basal melting and refreezing (e.g., Yan et al., 2025b), evaluate how debris entrainment affects basal ice rheology (e.g., Rempel et al., 2023), and predict radar attenuation based on modeled englacial temperature fields (e.g., MacGregor et al., 2007). Such observational, modeling, and sampling work would provide a powerful framework for testing competing basal unit formation mechanisms and improving our understanding of basal ice processes.

## 5. Potential geological control on basal thermal condition

We observe a strong correlation between the presence and fractional thickness of incoherent scattering and the subglacial topographic roughness, defined as the standard deviation of bed elevation over a 400-meter horizontal window (Fig. 2-d). Above the rugged terrain of the Subglacial Gamburtsev Mountains, where topographic roughness is high, we observe a correspondingly high fractional thickness of incoherent scattering within the basal unit. As the ice flows downstream into the relatively smooth South Pole Basin, the fractional thickness of scattering decreases and eventually disappears. Further downstream, as the ice approaches the Recovery Subglacial Highlands—where topographic roughness again increases—incoherent scattering re-emerges, with fractional thicknesses exceeding 90% in some areas.

It is possible that variation in subglacial geology exerts a primary control on both basal thermal conditions and subglacial roughness, thereby driving the observed correlation between incoherent scattering and bed topography. In particular, geological heterogeneity—especially when coupled with the presence of subglacial water—may redistribute the background geothermal flux, leading to elevated basal temperatures in localized areas and enhancing radar signal attenuation (Yan et al., 2022a). At the same time, contrasts in lithology and tectonic structure can influence patterns of erosion and sediment deposition, shaping the subglacial landscape and its roughness (Yan et al., 2022b). Together, these processes suggest that the spatial variability of basal unit radar signature may reflect a coupled system in which subglacial geology governs both the basal thermal regime and subglacial landform.

This interpretation remains a quantitative hypothesis that requires further validation. Ongoing work within NSF COLDEX is investigating the subglacial geological and hydrological conditions of the region using IPR sounding and potential field datasets (Kerr et al., 2023, 2024). Follow-up modeling work can build on these constraints to simulate englacial temperature fields and estimate corresponding radar attenuation profiles. Comparing these modeled attenuation patterns with radar observations would offer a critical test of whether the observed transitions in basal boundary type and scattering characteristics can be attributed to thermally driven variations in radar signal propagation. Such work is also essential for assessing the potential of radar-derived basal unit characteristics as indirect indicators of basal thermal structure.

## 6. Impact of elevated noise floor

We observe an elevated noise floor in radar data from several flight lines during the survey (Fig. 8). Although this does not compromise overall data quality for mapping major features like bed topography or thick internal layers, it does hinder the identification of weaker, diffuse features such as incoherent scattering. In affected transects, higher background noise reduces the contrast needed to visually detect and map basal scattering. To illustrate this effect, Fig. 8 compares intersecting flight lines with differing noise levels, highlighting how noise conditions impact the visibility of incoherent scattering.

Additionally, we notice noise arising from electromagnetic interference (EMI) between the MARFA and UHF radar systems. Examples of such EMI noise are visible in Fig. 4a near the 100, 250, and 450 km distance marks at two-way travel times deeper than 35 μs, and in the right-side panel of Fig. 8-a and Fig. 8-e. The EMI noise appears to impact the delay–Doppler

analysis by producing spurious specular returns, which interfere with and obscure the real radar signal. The EMI was remedied midway through the first survey season (2022–23), so only the earliest transects from the first season are affected.

These observations underscore an important consideration for future surveys targeting fine-scale features: while data may appear high quality in general terms, reliable mapping of low-contrast structures depends heavily on signal-to-noise performance. System sensitivity, signal processing strategies, EMI mitigation between radar systems, and noise control all play critical roles in reliable radar-based detection. Therefore, the competency and configuration of radar systems—particularly for deep-ice sounding—must be carefully considered when designing surveys or interpreting mapping results.

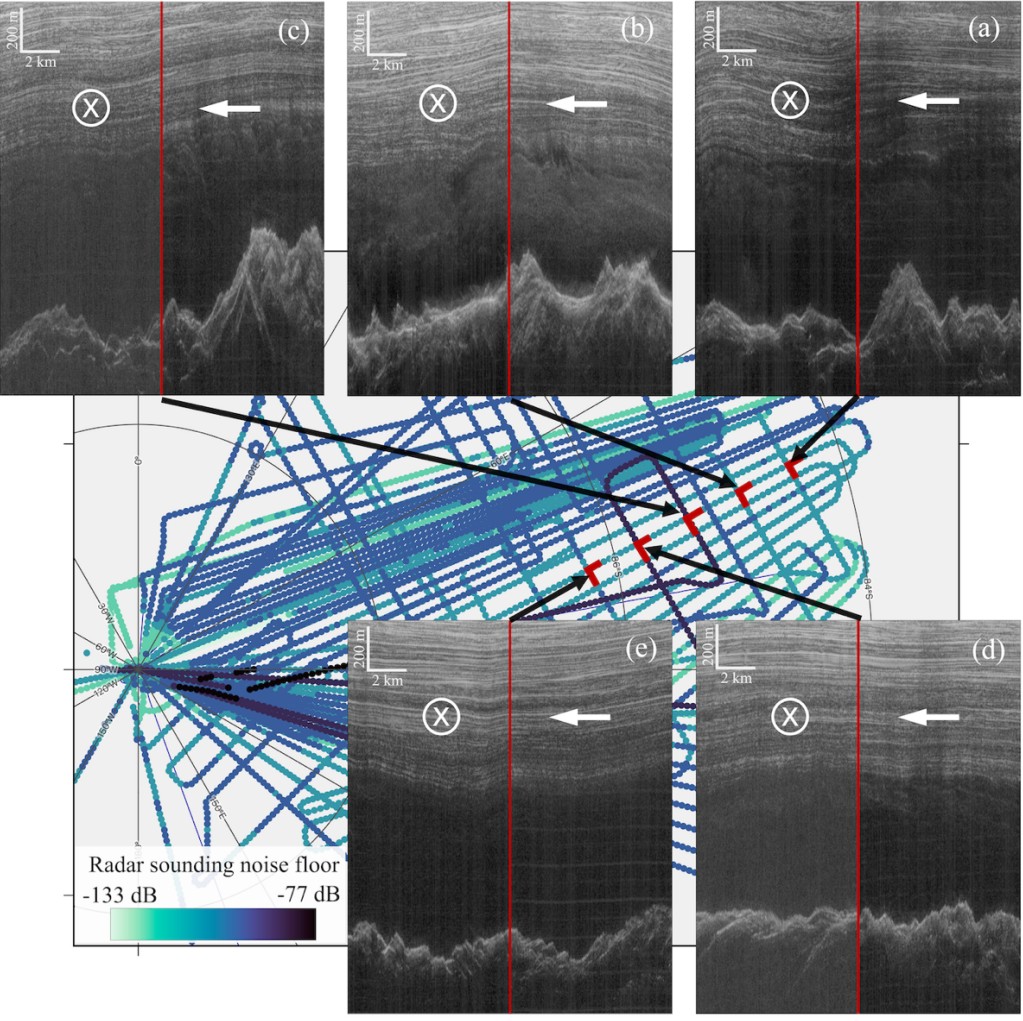

**Figure 8. Comparison of basal unit appearance at the intersection point of intersecting radar transects, illustrating the impact of elevated noise floor. Survey lines are color-coded by noise floor, with darker colors indicating higher noise levels. Radargrams from the intersecting transects are shown to demonstrate how elevated noise reduces the visibility of incoherent scattering within the basal unit. This map covers the same area as Fig. 2 and Fig. 6. The noise floor of each shown radargram at the intersection**
**point is: (a) left: -116 dB, right: -115 dB; (b) left: -117 dB, right: -118 dB; (c) left: -107 dB, right: -115 dB; (d) left: -106 dB, right: -115 dB; (e) left: -114 dB, right: -115 dB.**

## 7. Conclusion

This study leverages new ice-penetrating radar data from the NSF COLDEX airborne geophysics campaign to investigate the basal unit along the southern flank of Dome A, East Antarctica. Through manual mapping and Delay-Doppler analysis, we document the spatial variation of incoherent scattering within the basal unit and identify two types of basal unit top boundary: a sharp specular-to-scattering transition (type I) and a gradual attenuation-driven disappearance of specular stratigraphic reflections (type II). Our results show that incoherent scattering is most prevalent upstream near Dome A and diminishes downstream as ice flows towards the South Pole, a trend that correlates with both subglacial topographic roughness and shift from type I to type II boundary types.

We interpret this trend as a result of spatial variability in englacial temperature, with warmer ice in the downstream region increasing radar attenuation and suppressing the visibility of deep reflections. This interpretation is further supported by the consistent disappearance of incoherent scattering from the base upward. Moreover, the observed correlation between incoherent scattering and subglacial roughness may point to underlying geological controls, in which subglacial lithology influences both basal temperature and subglacial landform. Together, these interpretations highlight the need for future investigations—through numerical modelling, targeted sampling, and in situ measurements—to better constrain englacial temperature fields and subglacial geological conditions.

## 8. Data Availability

Unfocused IPR sounding data can be accessed at https://doi.org/10.15784/601768. Focused IPR sounding data can be accessed through the Open Polar Radar GeoPortal at: https://data.cresis.ku.edu/data/rds/2022_Antarctica_BaslerMKB/ and https://data.cresis.ku.edu/data/rds/2023_Antarctica_BaslerMKB/. IPR measured subglacial topography, surface elevation, subglacial roughness, and subglacial specularity content can be found at: https://doi.org/10.18738/T8/M77ANK. The thickness variation of the basal unit can be found at https://doi.org/10.15784/601912. Fractional thickness of incoherent scattering within the basal unit can be found at: https://doi.org/10.15784/601972. Delay-Doppler analysis result can be found at: https://dataverse.tdl.org/previewurl.xhtml?token=b81c2f4c-6f76-4532-9476-05ff303debb2.

## 9. Author Contribution

D.Y., S.S., and M.K. participated in field data acquisition, with S.Y. and D.B. contributing to the design of the field survey. Manual mapping of radar features was conducted by S.Y., A.V.-G., and S.S. D.Y. led the Delay-Doppler analysis. Figures were prepared by S.Y., D.Y., and D.L. All authors contributed to data interpretation and manuscript writing and approved the final version of the paper.

## 10. Competing Interests

The authors declare that they have no conflict of interest.

## 11. Acknowledgements

This work was supported by the NSF Center for Oldest Ice Exploration, an NSF Science and Technology Center (NSF 2019719), as well as the G. Unger Vetlesen Foundation. We thank the NSF Office of Polar Programs, the NSF Office of Integrative Activities, University of Texas at Austin, University of Washington, and Oregon State University for financial, logistical, and administrative support, and the NSF Antarctic Infrastructure and Logistics Program, Kenn Borek Air, Earthscope and the Antarctic Support Contractor for logistical support. We acknowledge the support of this work by Landmark Software and Services, a Halliburton Company. Maps in this manuscript were prepared using the QGIS platform, the Generic Mapping Tools (GMT, Wessel et al., 2019), and the Norwegian Polar Institute's Quantarctica package. This is UTIG contribution # 4145.

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
