# Peer review of "Basal Unit Radar Characteristics at the Southern Flank of Dome A, East Antarctica"

_EGUsphere, 2025_

## Referee Comment (RC1)

Yan et al. investigate the basal unit of the central Antarctic Ice Sheet, a critical yet poorly understood component of the ice-sheet system. They use new high-resolution radar data to examine the reflection and scattering properties in this region. Through Delay-Doppler analysis, they characterize two areas with different scattering properties. These results are used and discussed to determine possible englacial causes for the differences observed in these two regions, with variations in ice temperature and subglacial geology likely being responsible.

This paper is very well written, well structured, and addresses a relevant aspect of the Antarctic Ice Sheet. The basal unit is a region where complex and poorly understood processes lead to the loss of englacial layering and coherence in radar signals. Yan et al. provide a clear and comprehensive introduction to this complex topic, thoroughly describe the survey region, data, and methods, and make a valuable contribution to the field through their interpretation and critical discussion. In my opinion, this article is of high scientific quality, sheds light on a significant topic, and presents new insights, making it relevant for *The Cryosphere*.

I have two main points that need clarification and a few minor suggestions that could make the paper clearer. Once these points are addressed, I look forward to seeing this article published in *The Cryosphere*.

**Main points**

1)

L220 - 223: *"During this freeze-on process, debris may become entrained in the ice, potentially contributing to the incoherent scattering observed in IPR sounding. Given the similar radar signature, we infer that the incoherent scattering observed in the upstream portion of the COLDEX survey region may have formed through the same mechanism."*

I would disagree with the authors' claim that the radar signature observed is "similar" to those in studies analyzing the entrainment of englacial debris. My comment is based on a purely visual comparison of the radargrams in this study (e.g., Figures 1, 3, 7) with those shown below in studies addressing entrained basal debris in the basal ice unit: Bell et al. (2011), Wrona et al. (2018), and Franke et al. (2023, 2024). When comparing the radar signatures in Figures 1, 3, or 7 of this study (where it is most visible) with those in the mentioned studies, I do not believe we are referring to the same signature. In my view, the signatures in Bell et al. (2011), Wrona et al. (2018), and Franke et al. (2023, 2024) exhibit (i) higher return power (sometimes comparable to the bed return power), (ii) are directly connected with the ice base, and (iii) are located in regions where subglacial freeze-on was modeled. I would argue that the signatures referred to in this study correspond to those seen, for example, in Figure 2 of Wrona et al. (2018) or Figure 2 of Franke et al. (2024), but located ***above*** the layer defined as sediment entrained.

This does not mean that the incoherent scattering detected in this study cannot originate from entrained particles, but it is not comparable to what the aforementioned studies refer to.

I leave it to the authors to decide how to address this comment and they might disagree. However, it should be clarified that—at least based on a visual comparison of the radargrams—we are not discussing the same phenomenon.

[Figure]

This paper: Figure 7 in Yan et al (2025; in review)

[Figure]

Figure 2 in Wrona et al. (2018)

[Figure]

| Figure 2 in Franke et al. (2024) | Figure 3 in Bell et al. (2011) |

**References**

Bell, R. E., Ferraccioli, F., Creyts, T. T., Braaten, D., Corr, H., Das, I., et al. (2011). Widespread Persistent Thickening of the East Antarctic Ice Sheet by Freezing from the Base. *Science*, *331*(6024), 1592–1595. https://doi.org/10.1126/science.1200109

Wrona, T., Wolovick, M. J., Ferraccioli, F., Corr, H., Jordan, T., & Siegert, M. J. (2018). Position and variability of complex structures in the central East Antarctic Ice Sheet. *Special Publications*, *461*(1), 113–129. https://doi.org/10.1144/sp461.12

Franke, S., Gerber, T., Warren, C., Jansen, D., Eisen, O., & Dahl-Jensen, D. (2023). Investigating the Radar Response of Englacial Debris Entrained Basal Ice Units in East Antarctica Using Electromagnetic Forward Modeling. *IEEE Transactions on Geoscience and Remote Sensing*, *61*, 1–16. https://doi.org/10.1109/tgrs.2023.3277874

Franke, S., Wolovick, M., Drews, R., Jansen, D., Matsuoka, K., & Bons, P. D. (2024). Sediment Freeze-On and Transport Near the Onset of a Fast-Flowing Glacier in East Antarctica. *Geophysical Research Letters*, *51*(6). https://doi.org/10.1029/2023gl107164

2)

The authors interpret that the cause for the appearance and disappearance of incoherent scattering in the basal unit is spatial variability in englacial temperature. They mention that other processes can also contribute, but warmer ice seems to be their main interpretation, and this is also very prominent in the abstract.

I wonder how robust this is or if other mechanisms, such as equally warm/cold ice with different degrees of internal mixing could be an equally likely cause. There is not much other data or information presented in the paper (basal reflectivity, estimates on englacial attenuation, …) that would support temperature differences to be the main cause (or I did overlook them). Maybe the authors could comment on that and (if possible) strengthen this interpretation, or consider to say that other mechanisms could be equally likely in relevant parts of the paper (e.g., abstract and conclusions).

Moreover, a connection between incoherent scattering and subglacial roughness is being made. Is there also a correlation between incoherent scattering and ice thickness?

**Minor points**

- I hope I didn't overlook this, but the paper does not provide the specific years of the radar flights. This should be pointed out more clearly in the abstract and data and methods section.
- Would it be possible to draw the dashed lines you have in Figure 3, also in the radargrams in Figure 4?
- Figure 5: Ice thickness is reducing from right to left in the conceptual figure. From your radargrams and bed topography map it seems that ice thickness is however fairly constant and probably even thicker in your Type II boundary. Maybe this could be addressed or corrected to avoid conclusions drawn from this figure (e.g., that the Type II region has thinner ice).
- L129: *[… 2D focusing was applied following the procedure described in Peters et al. (2007), which helps correct for off-nadir scattering… ]* – I believe this refers to along-track off-nair scattering and not cross-track off-nadir scattering?
- L53-58: Franke et al. (2024) could be cited here as well with regard to freeze-on of subglacial water and sediment entrainment at the onset of Jutulstraumen Glacier, but I'll leave it entirely optional for the authors because it concerns one of my own publications.
- L64: Regarding radar signatures and COF, I'd suggest to cite the work of Lilien et al. (2021) as well.
- L220: Regarding freeze-on of subglacial water, I'd suggest to cite Creyts et al. (2014) next to Wolovick et al. (2013) as well.

**References**

Creyts, T. T., Ferraccioli, F., Bell, R. E., Wolovick, M., Corr, H., Rose, K. C., Frearson, N., Damaske, D., Jordan, T., Braaten, D., and Finn, C.: Freezing of ridges and water networks preserves the Gamburtsev Subglacial Mountains for millions of years, Geophys Res Lett, 41, 8114–8122, https://doi.org/10.1002/2014gl061491, 2014.

Lilien, D. A., Steinhage, D., Taylor, D., Parrenin, F., Ritz, C., Mulvaney, R., Martín, C., Yan, J.-B., O'Neill, C., Frezzotti, M., Miller, H., Gogineni, P., Dahl-Jensen, D., and Eisen, O.: Brief communication: New radar constraints support presence of ice older than 1.5 Myr at Little Dome C, Cryosphere, 15, 1881–1888, https://doi.org/10.5194/tc-15-1881-2021, 2021.

Franke, S., Wolovick, M., Drews, R., Jansen, D., Matsuoka, K., and Bons, P. D.: Sediment Freeze-On and Transport Near the Onset of a Fast-Flowing Glacier in East Antarctica, Geophys. Res. Lett., 51, https://doi.org/10.1029/2023gl107164, 2024.

Thank you once more for the nice read and I hope to see this paper published soon.

Best wishes,
Steven Franke

---

## Referee Comment (RC2)

Review of https://doi.org/10.5194/egusphere-2025-3944: 'Basal Unit Radar Characteristics at the Southern Flank of Dome A, East Antarctica', by Yan et al.

This article describes the englacial basal units classified from the analysis of scattering in radar images. The authors discuss two types of units, depending on how sharp or gradual is the scattering variation, and relate the transition spatial transition between units to changes in englacial temperature and bed geomorphology. From my point of view, the main contributions of the article are the methods for limiting the upper layer of the basal unit, and for classifying the basal content with the two discussed types.

The structure of the paper facilitates the reading. In particular, the introduction nicely explains what the reader will find in the article. The images are very well chosen and meaningful, with scientific colour maps for universal interpretation. The references I checked support the article. The data are in public repositories and are open, except maybe one data set, maybe currently under embargo.

Below I include my comments and suggested additions and minor corrections. My main comments are related to how reflections are classified between specular or scattering, to the interpretation of the two discussed types of basal unit types, and to the presence of sidelobes in the SAR images. Please do not hesitate to disagree with any of my comments.

**Main comments**

- 1. The document is very well written, and very interesting to read. I personally found the reading very fluent. However, the next times I read the document, I found it more difficult to follow in detail. These are my reasons:
  - a. the names of the two reflection types ('specular', and 'scattering') and the 'incoherent' scattering. The term 'scattering' I think is quite general, and it could also include 'specular' scattering. For example, you also use the incoherent scattering to show the presence of the basal unit. I would change the term 'scattering' as a reflection type. In line 140-141 you explain the nature of the scattering of this type, as 'volume scatterer distributing the energy over a broad range of angles', and in line 160 with 'mode diffuse'. I think these two explanations are very clear to the reader, so maybe the new name could be obtained from there.
  - b. there are terms like 'true scattering energy' (line 154) and 'coherent scattering', that could be defined. For example, in the article I interpret that there is coherent scattering when the layers can be traced in the image, although I don't fully agree.
  - c. I was confused between the two types of basal unit content (type I and type II), and the two reflection types (specular and scattering). In my first reading, I thought they were the same, but later I realised that they were not. Maybe this could be made explicit in the text.
- 2. The introduction section 1.2 is great. In particular, the five mechanisms explaining why the basal unit can be more obscured, strongly catch the attention of the readers. However, only mechanisms 2) and 4) are included in the conclusions later in the article; for example the fabric is later mentioned only in line 155, and debris in section 4). I would mention which of these mechanisms are going to be discussed, and why some seem not that important (or at least this is my interpretation from the article). My main concern is that if mechanism 5) is not discussed, and 1) and 3) not seem important, maybe the attribution of basal units to englacial temperature and geomorphology could be not valid. For example, from my experience the polarization used (co-

- polar or cross-polar) affects the aspect of the basal unit: in cross-polar we can reach deeper, and better trace the basal-unit layers than with the co-polar.
- 3. The SAR images in Fig. 3 have sidelobes, or ghosts from the surface. They are not very clear in Fig. 3, but in Fig. 4a they affect the power ratios in travel times beyond 35 us, at horizontal distances ~100 km, ~250 km, and ~450 km. I wonder if Fig. 4b would be different without these sidelobes or ghosts. I mention this as a main comment because these two images are very important in the article, and I recognise that improving the SAR image in Fig. 3b could be challenging. If it can't be corrected, I would explain this for the reader to be aware.

**Minor comments**

- Line 47: with 'disrupted to ice flow', do you mean the base unit to be stagnant or faster?
- Line 68: does 'folding below range resolution' mean that the folding would be resolved in the image and then appear as incoherent?
- Figure 1 and Figure 3: I guess they are images with same polarization in transmission and reception, but I would mention it. In Fig. 1 there is a clear obscure banding within 100 km and 225 km; if it means a different ice fabric, maybe it could be commented for Fig. 3.
- Figure 2: I would increase the font size of the 200 m intervals.
- Line 119: the mapping is done manually or by the software package?
- Line 123: instead of 'same englacial reflection' I would write 'same layer'.
- Lines 120 and 130: I would check the references 'Yan et al....', because they might correspond to data sets, and referring to the 'Fig.3' and 'Fig 2-c' of those references could be not correct.
   From lines 425 to 430, the references 2022b and 2025a are in the same paragraph.
- Figure 3: very nice images, with the dash lines very helpful.
- Figure 4: I would include the same dash lines are in Fig.3 b.
- Line 143-147: to determine if there is a reflection in the output window, I guess there is a threshold in the amplitude levels. From my experience, I find this is not as trivial as it might seem, so it would be great to know your interpretation.
- Line 163: (this might be a very personal interpretation, so please ignore if you don't agree) because of the sharp transition in type I, maybe is better to remove in line 160 the word 'more' next to diffuse? 'more' might mean a transition compared to the upper layers.
- Line 190-192: I would mark in Fig. 3b and Fig. 4 the location at where the incoherently scattering disappears, and where the 'scattering diminishes' and 'vanishes at the bottom', because I'm not able to locate these descriptions in the images.
- Line 204-205: I'm confused with the expressions 'lowering the depth' and 'raising the ...
  depth'. The 'lowering' means deeper or shallower? I would remove the word 'attenuation',
  because the critical depth does not depend only on the attenuation, but on the amplitude.
- Line 217: with 'give rise to incoherent scattering', do you mean 'the erasing of the heterogeneities'? It might be interpreted as 'heterogeneities that rise incoherent scattering'.
- Line 227: I would add that 'radar data alone' cannot confirm the origin of the scattering reflectors, to reinforce what you say in lines 223-225.
- Section 5: the last paragraph is excellent, suggesting exploitation of legacy data and new models considering the radar data.
- Figure 7: it is very illustrative! I would just add the noise at each panel. My doubt is always whether these differences in perpendicular tracks are due to noise or to clutter.
- Data availability: for the data in Open Polar Radar, if possible, I would give a specific link to the data set. The Delay-Doppler analysis result is not currently open; is it under embargo?
- References: if there are references with only data sets, I think the term [data set] should be included.

---

## Author Comment (AC1)

Dear Dr. Franke,

We thank you for the time and effort in reviewing our manuscript. We sincerely appreciate the positive feedback and insightful comments, which have been incorporated into the revised manuscript. Below, we provide a detailed, point-by-point response to your comments. For each point, we reproduce the referee's comment in italics, followed by our response and a description of the corresponding changes in the manuscript. We believe these revisions have substantially improved the manuscript, and we hope you agree that it is now suitable for publication in The Cryosphere.

Sincerely,

Shuai Yan, on behalf of all authors
* * *
*Yan et al. investigate the basal unit of the central Antarctic Ice Sheet, a critical yet poorly understood component of the ice-sheet system. They use new high-resolution radar data to examine the reflection and scattering properties in this region. Through Delay-Doppler analysis, they characterize two areas with different scattering properties. These results are used and discussed to determine possible englacial causes for the differences observed in these two regions, with variations in ice temperature and subglacial geology likely being responsible.*

*This paper is very well written, well structured, and addresses a relevant aspect of the Antarctic Ice Sheet. The basal unit is a region where complex and poorly understood processes lead to the loss of englacial layering and coherence in radar signals. Yan et al. provide a clear and comprehensive introduction to this complex topic, thoroughly describe the survey region, data, and methods, and make a valuable contribution to the field through their interpretation and critical discussion. In my opinion, this article is of high scientific quality, sheds light on a significant topic, and presents new insights, making it relevant for The Cryosphere.*

*I have two main points that need clarification and a few minor suggestions that could make the paper clearer. Once these points are addressed, I look forward to seeing this article published in The Cryosphere.*

**Response**: We sincerely appreciate the positive feedback and insightful comments, which have led to substantial improvements in the manuscript.

*Main points:*

*1)L220 - 223: "During this freeze-on process, debris may become entrained in the ice, potentially contributing to the incoherent scattering observed in IPR sounding. Given the similar radar signature, we infer that the incoherent scattering observed in the upstream portion of the COLDEX survey region may have formed through the same mechanism."*

*I would disagree with the authors' claim that the radar signature observed is "similar" to those in studies analyzing the entrainment of englacial debris. My comment is based on a purely visual comparison of the radargrams in this study (e.g., Figures 1, 3, 7) with those shown below in studies addressing entrained basal debris in the basal ice unit: Bell et al. (2011), Wrona et al. (2018), and Franke et al. (2023, 2024). When comparing the radar signatures in Figures 1, 3, or 7 of this study (where it is most visible) with those in the mentioned studies, I do not believe we are referring to the same signature. In my view, the signatures in Bell et al. (2011), Wrona et al. (2018), and Franke et al. (2023, 2024) exhibit (i) higher return power (sometimes comparable to*

*the bed return power), (ii) are directly connected with the ice base, and (iii) are located in regions where subglacial freeze-on was modeled. I would argue that the signatures referred to in this study correspond to those seen, for example, in Figure 2 of Wrona et al. (2018) or Figure 2 of Franke et al. (2024), but located above the layer defined as sediment entrained.*

*This does not mean that the incoherent scattering detected in this study cannot originate from entrained particles, but it is not comparable to what the aforementioned studies refer to.*

*I leave it to the authors to decide how to address this comment and they might disagree. However, it should be clarified that—at least based on a visual comparison of the radargrams—we are not discussing the same phenomenon.*

**Response:** We appreciate your detailed and thoughtful comments here. In the figure below, we provide side-to-side comparisons of the COLDEX survey (used in this study) and the AGAP survey (used by Bell et al., 2011, and Wrona et al., 2018) at some of their intersection points:

[Figure]

[Figure]

Based on visual comparison, we agree that the incoherent scattering observed in the COLDEX data does not resemble the basal freeze-on packages shown in Bell et al. (2011) and Wrona et al. (2018). An example of such basal freeze-on packages is highlighted by the yellow arrow in the figure. Instead, the incoherent scattering indeed closely resembles the radar signature of the layer overlying the sediment-entrained unit.

Based on this observation, we interpret the incoherent scattering as arising from either (1) deformation and folding caused by ice flowing across alternating slippery and rough frozen patches of the bedrock (as suggested by Wolovick et al., 2012), or (2) variations in ice crystal orientation fabric (as suggested by Mutter and Holschuh, 2025).

We have revised the manuscript to include this comparison and changed our interpretation, which can be found at line #202-212 of the revised manuscript (line #264-278 of the track-change version). Specifically, we state: "The COLDEX survey is situated directly downstream of the Antarctica's Gamburtsev Province (AGAP) Project (Corr et al., 2020). It has been hypothesized that the AGAP IPR sounding reveals packages formed by freezing of subglacial water and subsequent entrainment of debris (Creyts et al., 2014; Wolovick et al., 2013). We provide side-by-side comparisons of this basal unit as imaged by the COLDEX and AGAP IPR sounding at several intersection points in Fig. 7. We notice that (1) the incoherent scattering exhibits characteristics similar to the unit directly overlying the basal freeze-on package, and (2) this incoherent scattering is widespread within the AGAP survey in the region intersecting the COLDEX survey, that is, around and downstream of the area where widespread basal freeze-on was inferred by Bell et al. (2011). Based on this observation, we consider the incoherent scattering unlikely to represent the basal freeze-on package given its distinct radar signature. Instead, we interpret the incoherent scattering as arising from either (1) deformation and folding caused by ice flowing across slippery patches of the bedrock (as suggested by Wolovick et al., 2012), or (2) variations in ice crystal orientation fabric (as suggested by Mutter and Holschuh, 2025)."

We also note in the figure caption that: "the radar system used in the AGAP survey operates at a different center frequency (150 MHz), which results in different vertical resolution and may alter the appearance of the same reflector—particularly for reflectors whose characteristic dimensions are comparable to the radar wavelength."

*2) The authors interpret that the cause for the appearance and disappearance of incoherent scattering in the basal unit is spatial variability in englacial temperature. They mention that other processes can also contribute, but warmer ice seems to be their main interpretation, and this is also very prominent in the abstract.*

*I wonder how robust this is or if other mechanisms, such as equally warm/cold ice with different degrees of internal mixing could be an equally likely cause. There is not much other data or information presented in the paper (basal reflectivity, estimates on englacial attenuation, ...) that would support temperature differences to be the main cause (or I did overlook them). Maybe the authors could comment on that and (if possible) strengthen this interpretation, or consider to say that other mechanisms could be equally likely in relevant parts of the paper (e.g., abstract and conclusions).*

*Moreover, a connection between incoherent scattering and subglacial roughness is being made. Is there also a correlation between incoherent scattering and ice thickness?*

**Response:** Thanks for the detailed and thoughtful comments and suggestions. We have expanded the discussion of the other mechanisms that are outlined in the introduction section, which can be found at the Section 4 and Section 5 of the revised manuscript. Specifically, we discuss the freeze-on hypothesis at line #202-210 of the revised manuscript (line #264-274 of the track change version), leveraging the direct side-to-side comparison of the COLDEX and the AGAP radar survey. Following this discussion, we discuss the deformation and folding hypothesis at line #210-212 of the revised manuscript (line #274-278 of the track-change version), in addition to what was included in the original manuscript at line #231-239 of the revised manuscript (line #293-329 of the track-change version). The hypothesis around englacial temperature variation was already discussed extensively in the original manuscript.

We do not see a strong correlation between the ice thickness and the presence and the thickness variation of the incoherent scattering.

*Minor points:*

● *I hope I didn't overlook this, but the paper does not provide the specific years of the radar flights. This should be pointed out more clearly in the abstract and data and methods section.*

**Response:** We appreciate your suggestion. We have provided a clarification about the specific years when the surveys were conducted (2022-23 and 2023-24) at line #103 of the revised manuscript (line #131 of the track-change version).

● *Would it be possible to draw the dashed lines you have in Figure 3, also in the radargrams in Figure 4?*

**Response:** Thanks for the suggestion. We have added dash lines to Figure 4a to highlight the basal unit and incoherent echo.

● *Figure 5: Ice thickness is reducing from right to left in the conceptual figure. From your radargrams and bed topography map it seems that ice thickness is however fairly constant and probably even thicker in your Type II boundary. Maybe this could be addressed or corrected to avoid conclusions drawn from this figure (e.g., that the Type II region has thinner ice).*

Response: We have revised the figure caption to explicitly note that "the variations in ice thickness and subglacial topography shown in this conceptual sketch are intended only as a schematic illustration and do not necessarily correspond to actual correlations between such variations and basal unit boundary types."

● *L129: [... 2D focusing was applied following the procedure described in Peters et al. (2007), which helps correct for off-nadir scattering... ] – I believe this refers to along- track off-nair scattering and not cross-track off-nadir scattering?*

**Response:** We have made the corresponding correction at line #129 of the revised manuscript (line #170 of the track-change version).

● *L53-58: Franke et al. (2024) could be cited here as well with regard to freeze-on of subglacial water and sediment entrainment at the onset of Jutulstraumen Glacier, but I'll leave it entirely optional for the authors because it concerns one of my own publications.*

● *L64: Regarding radar signatures and COF, I'd suggest to cite the work of Lilien et al. (2021) as well.*

● *L220: Regarding freeze-on of subglacial water, I'd suggest to cite Creyts et al. (2014) next to Wolovick et al. (2013) as well.*

**Response:** Thank you for the suggestions. We have included these references in the revised manuscript.

---

## Author Comment (AC2)

Dear Dr. Pingarron,

We thank you for the time and effort in reviewing our manuscript. We sincerely appreciate the positive feedback and insightful comments, which have been incorporated into the revised manuscript. Below, we provide a detailed, point-by-point response to your comments. For each point, we reproduce the referee's comment in italics, followed by our response and a description of the corresponding changes in the manuscript. We believe these revisions have substantially improved the manuscript, and we hope you agree that it is now suitable for publication in The Cryosphere.

Sincerely,

Shuai Yan, on behalf of all authors
* * *
*This article describes the englacial basal units classified from the analysis of scattering in radar images. The authors discuss two types of units, depending on how sharp or gradual is the scattering variation, and relate the transition spatial transition between units to changes in englacial temperature and bed geomorphology. From my point of view, the main contributions of the article are the methods for limiting the upper layer of the basal unit, and for classifying the basal content with the two discussed types.*

*The structure of the paper facilitates the reading. In particular, the introduction nicely explains what the reader will find in the article. The images are very well chosen and meaningful, with scientific colour maps for universal interpretation. The references I checked support the article. The data are in public repositories and are open, except maybe one data set, maybe currently under embargo.*

*Below I include my comments and suggested additions and minor corrections. My main comments are related to how reflections are classified between specular or scattering, to the interpretation of the two discussed types of basal unit types, and to the presence of sidelobes in the SAR images. Please do not hesitate to disagree with any of my comments.*

**Response**: We sincerely appreciate the positive feedback and insightful comments, which have led to substantial improvements in the manuscript.

*Main comments*

*1. The document is very well written, and very interesting to read. I personally found the reading very fluent. However, the next times I read the document, I found it more difficult to follow in detail. These are my reasons:*

> *a. the names of the two reflection types ('specular', and 'scattering') and the 'incoherent' scattering. The term 'scattering' I think is quite general, and it could also include 'specular' scattering. For example, you also use the incoherent scattering to show the presence of the basal unit. I would change the term 'scattering' as a reflection type. In line 140-141 you explain the nature of the scattering of this type, as 'volume scatterer distributing the energy over a broad range of angles', and in line 160 with 'mode diffuse'. I think these two explanations are very clear to the reader, so maybe the new name could be obtained from there.*

*b. there are terms like 'true scattering energy' (line 154) and 'coherent scattering', that could be defined. For example, in the article I interpret that there is coherent scattering when the layers can be traced in the image, although I don't fully agree.*

*c. I was confused between the two types of basal unit content (type I and type II), and the two reflection types (specular and scattering). In my first reading, I thought they were the same, but later I realised that they were not. Maybe this could be made explicit in the text.*

**Response:** We appreciate you pointing out this mixed usage of terminology and the potential confusion it may cause. We have thoroughly revised the manuscript to standardize the terminology. We now use the term "non-stratigraphic, incoherent echo" to describe the feature previously referred to as "incoherent scattering," and we reserve the term "incoherent scattering" for cases after the delay–Doppler results confirm that the energy is indeed scattering. The only exception occurs in the part of the Introduction section where we refer to the study by Mutter and Holschuh (2025), which is prior to presenting our delay–Doppler results. There, we retain the term "incoherent scattering" to remain consistent with their original terminology.

The relationship and distinction between basal unit boundary types (type I vs. type II) and reflection types (specular vs. scattering) can be found at line #166-170 of the revised manuscript.

*2. The introduction section 1.2 is great. In particular, the five mechanisms explaining why the basal unit can be more obscured, strongly catch the attention of the readers. However, only mechanisms 2) and 4) are included in the conclusions later in the article; for example the fabric is later mentioned only in line 155, and debris in section 4). I would mention which of these mechanisms are going to be discussed, and why some seem not that important (or at least this is my interpretation from the article). My main concern is that if mechanism 5) is not discussed, and 1) and 3) not seem important, maybe the attribution of basal units to englacial temperature and geomorphology could be not valid. For example, from my experience the polarization used (co-polar or cross-polar) affects the aspect of the basal unit: in cross-polar we can reach deeper, and better trace the basal-unit layers than with the co-polar.*

**Response:** Thanks for the detailed and thoughtful comments and suggestions. We have expanded the discussion of the other mechanisms that are outlined in the introduction section, which can be found at the Section 4 and Section 5 of the revised manuscript. Specifically, we discuss the freeze-on hypothesis at line #202-210 of the revised manuscript (line #264-274 of the track-change version), leveraging the direct side-to-side comparison of the COLDEX and the AGAP radar survey. Following this discussion, we discuss the deformation and folding hypothesis at line #210-212 of the revised manuscript (line #274-278 of the track-change version), in addition to what was included in the original manuscript at line #231-239 of the revised manuscript (line #293-329 of the track-change version). The hypothesis around englacial temperature variation was already discussed extensively in the original manuscript.

*3. The SAR images in Fig. 3 have sidelobes, or ghosts from the surface. They are not very clear in Fig.3, but in Fig. 4a they affect the power ratios in travel times beyond 35 us, at horizontal distances ~100 km, ~250 km, and ~450 km. I wonder if Fig. 4b would be different without these sidelobes or ghosts. I mention this as a main comment because these two images are very important in the article, and I recognise that improving the SAR image in Fig. 3b could be challenging. If it can't be corrected, I would explain this for the reader to be aware.*

**Response:** The noise noted by the reviewer here is the result of electromagnetic interference (EMI) between the MARFA and UHF radar systems. This issue was remedied midway through the first survey season (2022–23), so only the earliest transects from that season are affected. We have noted this issue in Section 6 at line #294-304 of the revised manuscript (line #398-404 of the track-change version). Specifically, we now clarify: "Additionally, we notice noise arising from electromagnetic interference (EMI) between the MARFA and UHF radar systems. An example is visible in Fig. 4a near the 100, 250, and 450 km distance marks at two-way travel times deeper than 35 μs, and visible in the right-side panel of Fig. 8a and Fig. 8e. The EMI noise appears to impact the delay–Doppler analysis by producing spurious specular returns, which interfere with and obscure the real radar signal. The EMI was remedied midway through the first survey season (2022–23), so only the earliest transects from the first season are affected."

*Minor comments*

*– Line 47: with 'disrupted to ice flow', do you mean the base unit to be stagnant or faster?*

**Response:** Under this hypothesis, no special deformation profile is assumed for the basal unit relative to the stratigraphic ice above. Instead, the hypothesized disruption arises from ice flowing over rough subglacial terrain. Because the basal unit is in direct contact with this terrain, it is more susceptible to such disruption. We have added this clarification to the revised manuscript at line #47-50 (line #63-66 of the track-change version).

*– Line 68: does 'folding below range resolution' mean that the folding would be resolved in the image and then appear as incoherent?*

**Response:** This statement is referring to the study done by Mutter and Holschuh (2024), where they stated "Macro-scale deformation and layer folding at scales below the range resolution of radar do not seem to result in incoherent scattering or induce an echo-free zone as has been previously hypothesized." According to their analysis, such folding would not produce either the incoherent scattering or the echo-free zone that this study is investigating.

*– Figure 1 and Figure 3: I guess they are images with same polarization in transmission and reception, but I would mention it. In Fig. 1 there is a clear obscure banding within 100 km and 225 km; if it means a different ice fabric, maybe it could be commented for Fig. 3.*

**Response:** The noise noted by the reviewer here is the result of electromagnetic interference (EMI) between the MARFA and UHF radar systems. We have noted this issue in Section 6 at line #294-304 of the revised manuscript (line #398-404 of the track-change version).

*– Figure 2: I would increase the font size of the 200 m intervals.*

**Response:** Thanks for your suggestion! A corresponding change has been made to Figure 2.

*– Line 119: the mapping is done manually or by the software package?*

**Response:** As noted in the manuscript at line #122, the software supports semi-automatic tracing with user input. The user needs to manually mark some key points outlining the feature of interest, and the algorithm will then complete the trace by tracking the local maximum of received echo power.

*– Line 123: instead of 'same englacial reflection' I would write 'same layer'.*

**Response:** We appreciate the suggestion, but we respectfully argue that "reflection" is the more precise term here, as "layer" can be misinterpreted as an ice layer between two reflections (i.e., ice unit).

*– Lines 120 and 130: I would check the references 'Yan et al....', because they might correspond to data sets, and referring to the 'Fig.3' and 'Fig 2-c' of those references could be not correct. From lines 425 to 430, the references 2022b and 2025a are in the same paragraph.*

**Response:** The figure references here (Fig. 3 and Fig. 2-c) refer to figures in this manuscript. We apologize for the confusion.

*– Figure 3: very nice images, with the dash lines very helpful.*

*– Figure 4: I would include the same dash lines are in Fig.3 b.*

*– Line 190-192: I would mark in Fig. 3b and Fig. 4 the location at where the incoherently scattering disappears, and where the 'scattering diminishes' and 'vanishes at the bottom', because I'm not able to locate these descriptions in the images.*

**Response:** Thanks for the suggestion. We have added dash lines to Figure 4a to highlight the basal unit and the incoherent scattering echo. We also expanded the description of the incoherent-scattering echo's thickness variation: "This fraction gradually decreases downstream as the ice flows toward the South Pole Basin, and eventually, the incoherent scattering disappears entirely and the basal unit manifests solely as an echo-free zone (Fig. 3, Fig. 6-b). Notably, during this transition from full scattering to entirely echo-free, the scattering consistently diminishes from the base upward—i.e., the echo-free zone first develops at the bottom of the basal unit, immediately above the bedrock, and then progressively thickens upward as it evolves downstream (Fig. 3)." (line #195-200 of the revised manuscript and line #248-262 of the track-change version). We hope these revisions improve clarity.

*– Line 143-147: to determine if there is a reflection in the output window, I guess there is a threshold in the amplitude levels. From my experience, I find this is not as trivial as it might seem, so it would be great to know your interpretation.*

**Response:** Thanks for pointing this out. We have added further explanation on this point at line #146 of the revised manuscript (line #187 of the track-change version). Specifically, we now clarify: "In this study, we apply Doppler filtering using 1000-meter along-track apertures to compare the SNR of returns from three angular windows: nadir and ±11° off-nadir (in air), with evaluations spaced every 500 meters along the flight path. Energy that appears only in one angular view is classified as specular, while that which is observed in all views is classified as scattering. We use the gradient in the ratio of specular to scattering of 10 dB/μsec to identify the top of the basal unit and a 3 dB scattered/specular ratio threshold to identify englacial scattering below that limit."

*– Line 163: (this might be a very personal interpretation, so please ignore if you don't agree) because of the sharp transition in type I, maybe is better to remove in line 160 the word 'more' next to diffuse? 'more' might mean a transition compared to the upper layers.*

**Response:** We appreciate and agree with this suggestion. A corresponding change has been made.

*– Line 204-205: I'm confused with the expressions 'lowering the depth' and 'raising the ... depth'. The 'lowering' means deeper or shallower? I would remove the word 'attenuation', because the critical depth does not depend only on the attenuation, but on the amplitude.*

**Response:** Thanks for pointing out this potential confusion. We have changed the sentence to: "Additionally, subglacial melting in warmer areas may remove scattering reflectors from the

base of the basal unit, thereby shifting the remaining scattering reflectors to greater depths, while simultaneously raising the critical depth at which radar reflections fall below the noise floor." (line #222-225 of the revised manuscript and line #288-291 of the track-change version).

*– Line 217: with 'give rise to incoherent scattering', do you mean 'the erasing of the heterogeneities'? It might be interpreted as 'heterogeneities that rise incoherent scattering'.*

**Response:** Thanks for pointing out this potential confusion. Here we mean that the processes of diffusion and deformation may erase the dielectric contrasts responsible for scattering. We have changed the sentence to "Together, diffusion and deformation may progressively erase the dielectric contrasts responsible for scattering, leading to its gradual disappearance downstream." (line #239 of the revised manuscript and line #338 of the track-change version)

*– Line 227: I would add that 'radar data alone' cannot confirm the origin of the scattering reflectors, to reinforce what you say in lines 223-225.*

**Response:** Thanks for the suggestion. We have substantially revised this section, and this sentence now reads "The radar data we have so far cannot definitively resolve the causes of the absence of stratigraphic reflections and the presence and thickness variation of incoherent scattering within the basal unit." (line #247 of the revised manuscript and line #341 of the track-change version)

*– Section 5: the last paragraph is excellent, suggesting exploitation of legacy data and new models considering the radar data.*

**Response:** We appreciate your positive feedback!

*– Figure 7: it is very illustrative! I would just add the noise at each panel. My doubt is always whether these differences in perpendicular tracks are due to noise or to clutter.*

**Response:** Thanks for the suggestion and the positive feedback! We have noted the noise floor of each panel in the revised figure caption, which now reads: " Comparison of basal unit appearance at the intersection point of intersecting radar transects, illustrating the impact of elevated noise floor. Survey lines in the background map are color-coded by noise floor, with darker colors indicating higher noise levels. Radargrams from the intersecting transects are shown to demonstrate how elevated noise reduces the visibility of incoherent scattering within the basal unit. The noise floor of each shown radargram at the intersection point is: (a) left: -116 dB, right: -115 dB; (b) left: -117 dB, right: -118 dB; (c) left: -107 dB, right: -115 dB; (d) left: -106 dB, right: -115 dB; (e) left: -114 dB, right: -115 dB."

*– Data availability: for the data in Open Polar Radar, if possible, I would give a specific link to the data set. The Delay-Doppler analysis result is not currently open; is it under embargo?*

**Response:** Thanks for the suggestions. We have revised the Data Availability statement accordingly.

*– References: if there are references with only data sets, I think the term [data set] should be included.*

**Response:** Thanks for the suggestion. We have made corresponding changes in the revised manuscript.